# Globalization and the Transformation of Political Attitude Structures at the Party Level in the Arab World: Insights from the Cases of Egypt and Jordan

**Malek Abduljaber \* and Ilker Kalin** 

Light House Academic Services, LLC, Ann Arbor, MI 48103, USA; ilkerkaln@gmail.com
\* Correspondence: malikfayez@gmail.com

**Abstract:** In this paper, the outline, design, and findings of an ongoing research project on the effects of globalization on the transformation of political ideology in the Arab world at the political party level are presented. It is argued that globalization has altered the dimensionality, type, and structuration of political ideology in the Arab world. The structure of preferences among political actors in the region shifted from a unidimensional one in the post-independence era to become multidimensional in the contemporary period, defined by high rates of economic, cultural, and political globalization. Arab political parties no longer organize their platforms based on the Islamic–liberal, Islamist–secular, or cultural divides. An economic values-based dimension has emerged to divide party programs, adding a second, distinct and statistically independent dimension to the already existing classic church versus state cleavage. Further, a new family of Islamist parties has emerged due to the economic, cultural, and political gains from globalization. This project argues that globalization causes political ideological shifts in attitudes through formulating new groups, schedules of preferences, and political/economic opportunities. This research contributes to the ongoing debate on the influence of globalization and any other social transformation process on changing political actors' preferences across time and space.

**Keywords:** globalization; political ideology; Arab world

## 1. Introduction

Often, researchers have investigated the trans-national political consequences of globalization on world politics [1–11]. More recently, however, emerging literature in comparative politics has noted that globalization has had a great political impact at the domestic/national level [12–17]. Continuing in the footsteps of this research agenda, focus is placed on the influence of globalization on the national politics of Arab nations, specifically Jordan and Egypt. This project focuses on the cultural as well as economic manifestations of globalization, mainly including privatization, increasing migration, increasing cultural integration, and the denationalization of domestic politics in the Arab world. Simply put, the main goal is to explore the significance of globalization on the structure of political attitudes in the Arab world at the political party or actor level. In other words, has globalization led to a transformation in the makeup of political divisions and the ideological standing of the main actors in the political arena over the past few decades in the Arab world? A natural inquisition concerning the level of analysis with respect to political parties has arisen among some readers and the simple answer is that there is a dearth of available data sources on the mass public to assist in the testing of the effects of globalization on ideological changes in ordinary citizens in the region. This research develops a new dataset that measures parties' positions at different time periods to assess the role of globalization in transforming the ideological space among parties.

Globalization is a general trend of denationalization, the unbundling of national boundaries [18–21]. Globalization is by no means a new phenomenon [22–24]. However, the magnitude of the process accelerated in the 1980s and 1990s throughout the Arab world [22,23,25–27]. Held (2007) notes that globalization, in its present form, has surpassed many similar historical epochs quantitatively as well as qualitatively [13]. Taken together and using the structural approach of party formation, one may think of globalization as either a critical juncture or a gradual process affecting the basic social structures leading to party formation [28–32].

Lipset and Rokkan (1967) noted that the deep sociological, economic, cultural, and religious divisions in each society provide the basic foundations for the party system within it [33]. The resulting dimensions structuring political divisions are called cleavages. Each country has a specific number, type, and structure of cleavages that are shaped and influenced by its historical development [34–38]. It is assumed here that globalization has a great effect that is capable of influencing the issue divide structure in the Arab world. Kriesi (2012) found that globalization has transformed the cultural divide in Western Europe, where immigration policy has defined a new cultural divide in recent decades at the national level in Austria, Germany, the Netherlands, Switzerland, England, and France [39]. This project extends the analysis to the almost forgotten region, the Arab world, and investigates whether globalization has led to any meaningful transformation in the national divide structures in Jordan and Egypt.

This study is an ongoing project aiming to explore the political significance of globalization on national issue divides in the Arab world. The underlying idea, design, and findings are presented with regard to that end. In the subsequent section, the theoretical framework in which globalization is hypothesized to affect issue divides is outlined. It is suggested that globalization is capable of creating new issue divides, or altering existing ones, through a variety of mechanisms. The complex cultural and economic structures generated by denationalization of boundaries create political potentials ready to be taken up by political actors, parties in particular, for mobilization. Conditional upon their position with respect to existing or new issue divides, parties may alter their policy for strategic reasons, responding to the changes induced by globalization.

## 2. A Theoretical Note on the Influence of Globalization on Cleavage Structures

The globalization can influence the transformation of political ideologies through some exogenous means. First, globalization is believed to create new forms of competition among the citizens of a given country [40–42]. Second, it is argued that citizens take positions on old and new issues that are ushered in by increased market, cultural, ethnic, and global integration [43–47]. Third, established and emerging political parties incorporate the new issues and old material into their electoral programs in a desire to appeal to their old voters and new potential pool of advocates [34–36,48,49]. One may generally understand that these effects, in return, create winners and losers who oppose each other with respect to the national issue divide structure [6,15,50–52]. This logic posits that globalization is a product of exogenous factors beyond the direct influence of incumbent governments, as clarified below.

Globalization is the product of endogenous, as well as exogenous, forces [53]. First, incumbent governments in democracies may prefer more open economic and cultural spheres backed by their electorates and implemented through public and foreign policies. Simultaneously, global markets and cultural integration may be prescribed by foreign governments and international institutions. Local governments or incumbents may not prefer globalization for a variety of reasons; for example, protecting local cultures or economic structures. Nevertheless, under dire economic and political circumstances such as those exhibited in Middle Eastern nations during the 1970s and 1980s, Western institutions induced higher levels of globalization under the guise of political and economic stability, thus protecting foreign interests in the region. This logic concludes that globalization is exogenous, as in the case of the MENA region, and is advocated by the present analysis.

Henry and Springborg (2010) proposed a globalization dialectic, suggesting that Middle Eastern governments are not free in deciding on the extent to which their economies or cultures are open [54].

The authors argued that Arab governments have long protected their societies from any extensive form of global integration. Despite relative success in the post-independent era, the endogenous effect of globalization represented by the will and desire of local elites to open up their economies and cultures to the world has eroded, and the Washington Consensus institutions, World Bank, and the International Monetary Fund, with strong backing from the United States and Western European powers, forced Arab regimes to accept and implement economic restructuring programs [55]. While one may argue that local elites desired to protect their regimes and opted to invite international institutions to fix their economies [16,52], this logic stands weak against the ample qualitative and quantitative evidence confirming the fact that Western institutions compelled the Jordanian and Egyptian regimes to implement their economic and cultural agendas, entailing a more open market and cultural sphere.

One may view losers as those who perceive the eroding of national boundaries, removal of economic barriers, and the increase of cultural integration as threats to their psychological and physical wellbeing [56–60]. On the other hand, one may see winners as those who benefit from economic, cultural, or societal integration, and have their lives generally enhanced [56,57,61]. Winners and losers compose different groups and expectations [56,61]. Overtime, such sociological changes give rise to a new economic class benefiting from globalization and reduce the economic potential of other groups previously protected by the less globalized system.

Political scientists have proposed a plethora of mechanisms by which globalization generates opposing outlooks in each society along economic, cultural, and political lines [62–65]. The single underlying observation lies in the creation of differing groups, winners and losers, with distinct preferences and heterogeneous composition [56,61]. In economic terms, the deep structural economic problems faced by Middle Eastern states in the 1980s led these countries to implement an economic restructuring adjustment package outlined by the International Monetary Fund and World Bank [66]. These programs essentially removed state subsidies for basic commodities and reduced state employment, one of the main sources of employment for Egypt and Jordan, and implemented neoliberal economic deregulation in hopes of attracting foreign investment. These measures hit the protected sectors of the economy and assisted emerging large private enterprises, in some cases causing small- and medium-sized businesses to flourish. This environment created a division along the lines of winners and losers of the globalization process based on economic return: those who lost state subsidies, employment, and could not move up the economic ladders resisted the process, while those who gained profits, employment, and economic return supported the process [67,68].

On the cultural sphere, globalization increased ethnic, linguistic, cultural, and religious contact among the world's population, including the Arab world [69–71]. In Jordan, there has been a tremendous migration from Egyptian workers, Iraqi refugees, Syrian refugees, and a host of other Libyan, Yemeni, and Lebanese citizens for various economic, social, and political reasons [72–74]. In Egypt, the situation is no different where the center of cultural interaction in the Arab world has only intensified in qualitative and quantitative terms. This environment has led to either an internal or external outlook from these populations [75,76]. On the one hand, a sizable portion of the nation perceives this cultural opening as an existential threat to its distinct pure Jordanian or Egyptian identity. On the other hand, supporters of this cultural opening rally around Pan-Arabism, Pan-Islamism, and more recently, cosmopolitanism. Parallel to the economic situation, this cultural environment creates polar opposites, and those who lose from the cultural interaction (perceived cultural loss, such as decreasing religiosity, and threats to national values) are viewed as losers. On the other hand, those who benefit from this cultural interaction on any level are seen as winners [77,78].

In political terms, globalization simply increases the interaction of state and non-state actors [79,80]. This increases the dissemination of political views, intensifies the existing ideological debates, and ignites citizen journalism, especially in the age of information technology [81,82]. This environment simply puts the dominant political divisions to the foreground. One important dimension in Arab politics is the regional race between Iran and Saudi Arabia and its political makeup. This political dynamic was best manifested in the aftermath of the Arab Spring where an increased

division intensified between those who supported the revolutions and those who did not. The political perception of gaining a victory or losing a privilege because of regional politics is one determinant of the winners and losers in political terms [83–85].

The logic presented in this analysis is consistent with the sociological view underlying political attitude and cleavage changes [86]. Social transformation processes such as modernization or globalization expose individuals and societies to new economic opportunities, lifestyles, political experiences, and cultural trends. With generation replacement, new voters along many old voters change their attitudes and outlooks towards politics, society, and religion [87]. Modernization theory is rich with explanations on how societies become less traditional with higher rates of urbanization and Westernization [88]. Globalization in this analysis is a social transformation process that gradually replaces old preferences with new ones. Such replacement does not necessarily entail the complete eradication of old values; on the contrary, a few sectors may become more vehement supporters of old traditions. Nevertheless, new preferences and more nuanced attitudes towards old and new issues emerge [89].

In sum, the winners of globalization seem to be those who benefit from neoliberal policies, small–medium sized enterprises, cultural cosmopolitans, and political advocates of integration. Meanwhile, the losers tend to be those who lost state support, being culturally inward looking and perceiving political losses from the process [56,61]. These issue divides generate political potentials for mobilization and articulation by political parties. The complexity of the political potential created by globalization coupled with the heterogeneous composition of winners and losers produces a difficulty for political parties to align with this potential given their traditional, established alignments. Further, this potentially opens up the political arena for new actors to organize and mobilize at the national level [56,57].

There are several moderators that facilitate the creation and change of political attitudes and their structures. First, prior levels of economic liberalization in the 1980s and 1990s moderate the relationship between globalization and ideological changes in the Arab World. Globalization creates winners and losers who formulate explicit expectations from potential governments and political institutions. Winners are more likely to desire increased economic opening in order to reap the fruits of the process. Note that not only economic benefits are incurred for some groups, but also social and cultural gains are gained by those who value and advocate for more liberal orders where Westernization and modernization define social orders in the Middle East. This creates the expectation that prior levels of economic and cultural openness moderate the relationship between globalization and political ideological structure change. In other words, when more open polities with established private markets and cultural hubs allowing for modernity exist, the expected change in political attitudinal structures in a polity is greater. In this case study, Egypt is expected to feel the change more than Jordan since Egypt has a larger private economic structure as well as a more vibrant cultural center, with closer ties to modernity and the West.

Further, political liberalization levels in Middle Eastern nation states moderate the relationship between globalization and political ideological structures. States featuring relatively free and fair elections in the 1980s are expected to have more vibrant civil societies that have experience in political campaigns and parliamentary life. This creates a conduit through which winners and losers funnel their expectations and platforms into existing parties or newcomers, since party entry is easier in more liberal regimes compared to more authoritarian ones. Therefore, we expect to observe the structural changes in political attitudes more readily in places where political liberalization has taken place in the 1980s and 1990s for Jordan and Egypt.

Third, cultural liberalization moderates the relationship between globalization and change in ideological structures. Prior cultural openness towards the West or the rest of the Arab or Islamic Worlds assists cultural winners from globalization in utilizing existing institutions to better disseminate their messaging. Cultural hubs like music, film, television production, print, and electronic presses can be used to distribute existing or new political content in order to make it more readily to the general public and political actors. Therefore, it is expected that in countries with established cultural

institutions, globalization's effect on political attitude changes would be observed less than in countries with lower cultural institution establishment.

The logic proposed here applies to both more and less liberal regimes. The difference lies in the level of liberalization. North America, Western Europe, and other well-known democracies exhibit a higher level of liberalization, political, cultural, and economic platforms. It is easier to establish a business entity, a cultural club, or a political party in the United States compared to Jordan. The labels "authoritarian", "semi-authoritarian", "hybrid", etc. for such groupings of political regimes simply refer to their liberalization levels. Such systems have less tolerance for political, economic, and cultural discourse and transactions, fearing the loss of control possessed by the elites of such nations. Therefore, it is expected to observe the change in political attitude structures in less advanced democracies more than established democracies because of low liberalization rates. Despite such a difference, the logic behind globalization's influence on political attitudes was originally developed in Western Europe and this analysis has extended it to the Arab World.

Figure 1 summarizes the logic of effects that globalization has on political preference change.

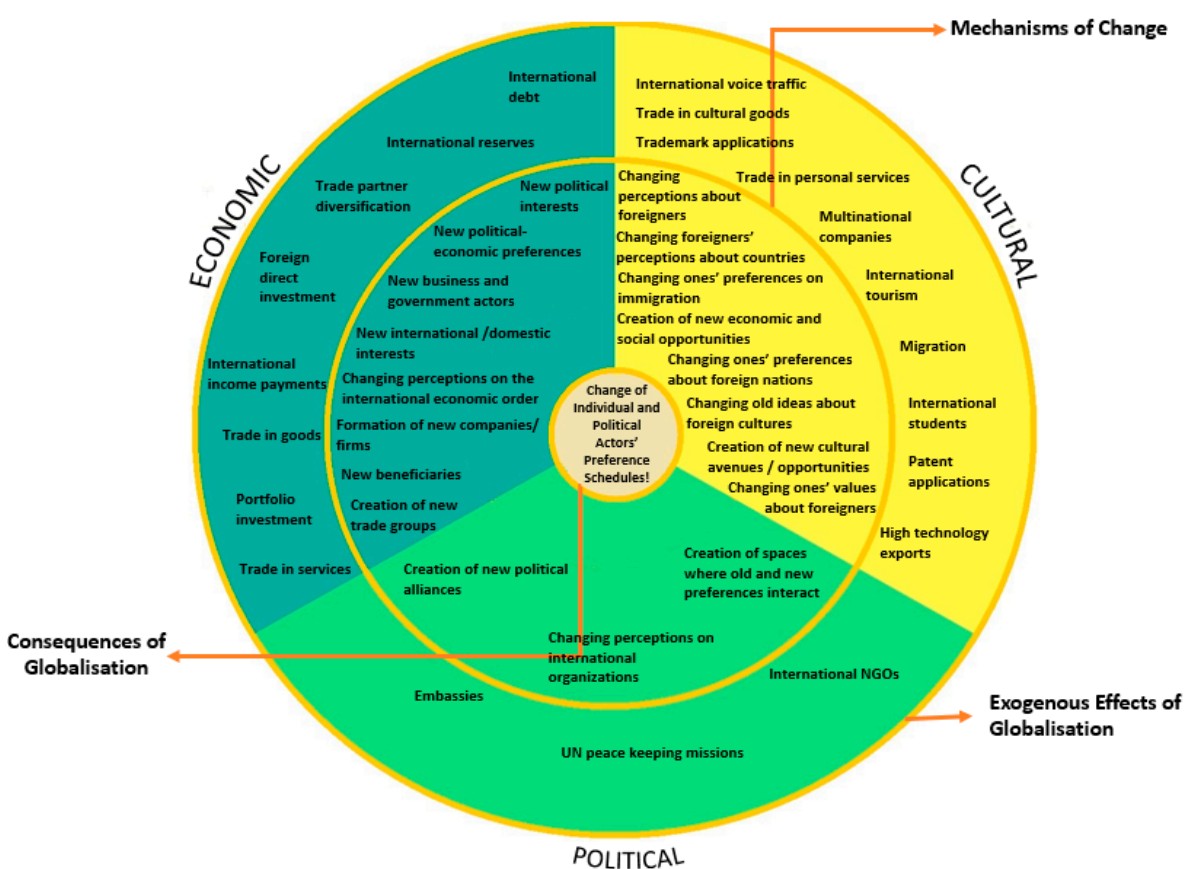

**Figure 1.** Conceptual model.

## 3. Research Design

To explore the influence of globalization on the cleavage structure in the Arab world two Arab states are analyzed: Egypt and Jordan. This choice was simply made due to data availability, quality, and accessibility. These countries are similar in many respects, but simultaneously differ on many measures [90,91]. On the one hand, the two cases feature hybrid regimes allowing for parliamentary elections and the Muslim Brotherhood and its off-shoots represent the main political opposition [92]. Further, Arabic is the main language and Islam is the dominant religion across the cases. On the other hand, the size of the economy, population, demographic makeup, political development of the state, and connection with the West are all different. Such similarities and differences make it difficult to

conduct a comparative method design such as most similar or different designs. Such designs are deterministic and do not provide clear evidence on the association or causality governing relationships among social or political variables [93]. This research is built on a quantitative research design based on an original dataset assembled from existing qualitative sources, particularly political documents.

To detect the effects of globalization on the structure of attitudes in any polity, one needs to measure political attitudes before and after globalization. Survey data in the Arab World prior to the Arab Barometer Project (2006) are rare and in many cases inaccessible [94]. Therefore, one needs to construct an original dataset measuring voters' positions on major political issues during the 1980s. One of the ways political scientists have done this is by measuring party positions by assuming that political parties highlight voters' concerns in their manifestos or leaders' speeches [95–97]. It would be absurd to assume that non-democracies, like Arab nation-states, are void of political attitudes or content. Like Western liberal democracies, Arab polities in the 1980s featured political actors, whether they supported the existing regime or opposed it. Newspapers and media outlets frequently reported such issues and actors' positions on them. This leaves researchers with no choice but relying on existing data sources, political documents, to measure parties' and voters' positions on relevant political content.

The present study's research design is framed within quantitative content analysis. The data for this project comes from political documents, party manifestos, and major newspapers within each country. Originally, such data comes in the form of sentences. Each sentence is taken separately and converted into a quantitative value based on a scoring scheme outlined below. To detect the effect of globalization, two periods are chosen for each country. One is before the country implemented the economic restriction program, the impetus for economic and cultural opening in the 1980s, and the other is following the "Arab Spring", the period with the most free and fair elections in the region's history. During this period, Arab political parties have formulated the lengthiest party programs outlining their visions and positions on political matters. Party programs prior to national parliamentary elections in each country are taken as the raw data measuring party positions. In the case of the first period where party programs were not fully developed, newspapers articles on parties and their positions on political issues are taken as the source of the data for this research.

A new dataset has been generated to meet the goals of this research. In order to identify the direction, salience, and position of parties with respect to the various issues at a given election, a content analysis of party manifestos was conducted. If manifestos were not found, secondary sources were used instead. For each election, two for each country, the manifestos of those parties that won seats in the house were analyzed. Using the Inter-Parliamentary Union data, parties' electoral programs for two elections per country have been analyzed (one prior to the intensified effect of globalization and the other before the most recent elections). For each manifesto, the general standing for the party on a given issue was sought.

Party manifestos and media reports of party positions prior to elections have been taken as objective sources for measuring parties' positions on political content. Party programs are distilled versions of the party standing on major and other issues important to the national electorate. It serves as the most authoritative document by which the party communicates its positions to voters. While parties and voters may not have the same ideological attitude structures and content, parties typically integrate voters' positions into their programs in an attempt to win votes. Therefore, the positions of various parties analyzed are representations of voter blocs within their polities. Similarly, media reports of party positions tend to deliver a transparent evaluation of each parties' positions on given political issues.

Kleinnijenhuis (2001) developed a method to analyze relationships between political objects (either political actors, or a political actor and a political issue) [98]. This paper only pertains to relationships between actors on the one hand, and actors and issues on the other. The method considers nine categories of major political content that political parties and voters have been concerned with across the Arab World and measures parties' positions on each. Every standing is assigned a value from $-2$ to $+2$, indicating the position of the political actor with seven intermediary possibilities ($-1.5$, $-1.0$, $-0.5$ and the same for positive standings). If the party is an advocate of an issue or policy relevant to

the issue, the position will be positive depending on the intensity of the position, and if it is against another issue or policy it will receive a negative number depending of the level of opposition.

An arising criticism of such a scoring method is that two parties may receive the same score in a given category, while one of them devotes a higher share of its manifesto to that issue compared to the other party. This analysis is interested in ideological structure at the party level, using their positions rather than the proportion of issues within their manifestos. For instance, Salafi parties may have short programs and call for the implementation of Islamic law more readily compared to Muslim Brotherhood (MB)-endorsed political parties that tend to have longer programs where Islamic law is referred to more often in comparison to Salafi parties. Given this, Salafi parties will be awarded a higher score on Islamic law compared to MB parties. The share of each issue in a party manifesto does not alter the associations of issues in a multidimensional space, the tool used to assess the dimensionality change before and after globalization.

The parties selected for the analysis received seats in the parliament, representing the various political spectrums in a single case, and reflecting the issues occupying the issue divide space. The parties selected in the analysis are:

- Jordan (1993): The Islamic Action Front (IAF), the Jordanian Communist Party (JCP), the Jordanian Ba'th Party (JBP), Al-Ahd/Yakatha (later known as the National Constitution Party, or NCP), and the National Union Party (NUP).
- Jordan (2013): The IAF, the Islamic Center Party (ICP), the JBP, the JCP, the NCP, the NUP, Stronger Jordan (SJ), Wifaq (Agreement), and the National Current Party.
- Egypt (1987): New Wafd, the Muslim Brotherhood (MB), the Labor Party, the Al Ahrar Party (Liberals), the National Democratic Party (NDP), and the Nationalist Progressive Unionist Party (NPUP).
- Egypt (2011): The Freedom and Justice Party (FJP, or MB representative), the Al Nour Party, the Islamic Center Party (Wasat), the Authenticity Party (AP), Revolution Continues (RC), New Wafd, the NPUP, the Social Democratic Party (SDP), the National Egyptian Party (the former members of the NDP), and the Freedom Party.

One common criticism on the validity of measures across studies of political attitudes and their structures concerns the choice of input metrics to the analysis [99,100]. The present study aims to estimate the ideological structure prior to and after globalization in the Arab World. This structure is composed of an infinite number of attitudes and factors. This analysis, therefore, only included nine of this infinite set; a limitation that cannot be escaped. The analysis mitigates such a weakness by only considering the relevant categories which voters and parties in the Arab World are concerned about. Few of them, such as economic and cultural liberalisms and religion and welfare, are common to all modern polities. Arab and Muslim world relations are peculiar to the region, and define many political debates across the modern 20th and 21st centuries' nation states in the region.

For political issues the detailed issue schema was revised and formulated as a simpler categorical framework that includes the following categories:

- Welfare: supports expansion of the welfare state and defense against welfare state entrenchment, tax reforms that have redistributive effects, employment programs, and health care programs.
- Budget: budgetary rigor, reduction of the state deficit, cut expenditures, and reduction of taxes that have no effect on redistribution of wealth.
- Economic liberalism (EconLib): support for deregulation, for more competition, and for privatization. Opposition to market regulation—this is the distinguishing criteria from the welfare category. Opposition to economic protectionism in agriculture and other sectors.
- Cultural liberalism (CultLib): support for the goals of the new social movements (with the exception of the environmental movement), peace, and solidarity with the third world, gender equality, and human rights. Support for cultural diversity, international cooperation, and for the United Nations. Opposition to racism, support for the right to abortion and euthanasia.

Opposition to patriotism, calls for national solidarity, defense of tradition, national sovereignty, and traditional moral values, and support for a liberal drug policy.

- Arab: support for Arab integration—including political and economic mergers.
- Palestine: support for peace with the State of Israel and normalization.
- West: support for increasing interaction with the United States and Western European countries.
- Religious influence (ReligionInf): support for the incorporation of religion into political, social, economic, and cultural spheres.
- Sharia: support for explicit implementation of Islamic law by the state apparatus.

The first three categories' refer to the classical economic dimension. Neoliberal outlooks expressing support for more privatization, less governmental control, and incentives for businesses lies at one end of the spectrum while a welfare vision supporting more income distribution, government social programs, and expanding the welfare state is on the other. The issues of cultural liberalism, sharia, and religious influence present the cultural dimension where liberals occupy one end (advocating for secular government, support for gender equality, and universal human rights) while opposing the more authoritarian view bringing religion into the political sphere. Finally, the issues of Palestine, the Arab world, and the West, reflect a third dimension concerned with whether the party espouses parochial or cosmopolitan trends toward international affairs such as regional integration, peace with Israel, and the relationship with the West.

This type of data allows us to construct visual maps showing party positions and issues in a low-dimensional space utilizing the multivariate technique of multidimensional scaling (MDS). MDS allows the researcher to represent the similarities and dissimilarities between objects in a flexible fashion [101]. For this study, the issue positions of parties provide the similarities and dissimilarities among parties and issues. For example, if an Islamist party supports the implementation of sharia, one would expect the distance between that party and category sharia to be small. If this information is represented in a common space, the distance and proximity of parties to issues supplies information regarding the structure of issue divides in a given country. MDS allows us to generate visual representations of the national issue divides, issue categories distances, and parties' positions along these issue divides.

Multidimensional scaling (MDS) is "a general methodology for producing a geometric model of proximities data" [102]. MDS is an exploratory statistical technique that generates a structure of a given dataset, the number of most interesting and meaningful dimensions latent in the observed dataset. The application of MDS is not linked with prior assumptions about the number of dimensions, relationships, and associations of variables similar to other techniques like structural equation modeling or confirmatory factor analysis. The input data for any MDS analysis involve a similarity or dissimilarity matrix, measuring distances between items or objects. MDS attempts to represent the distances in a dimensional space with a given number of dimensions, here the number of political factors defining the ideological space at the political party level. The number of dimensions is determined by the researcher in order to provide sound theoretical justification supporting the supplied interpretation.

One important feature offered by MDS is the removal of unnecessary assumptions regarding the structure of issue divides in Jordan and Egypt. Most analyses of issue divides make theoretical assumptions regarding the number, content, and association of the dimensions structuring the political debate among actors. Here, MDS allows the testing of the dimensionality and issues composing these dimensions without any assumptions.

## 4. Results

The developers of the KOF Index of Globalization conceptualize globalization as "the process of creating networks of connections among actors at multi-continental distances, mediated through a variety of flows including people, information and ideas, capital and goods. Globalization is conceptualized

as a process that erodes national boundaries, integrates national economies, cultures, technologies and governance and produces complex relations of mutual interdependence" [103] Figures 1 and 2 display the KOF Globalization Index scores for Jordan and Egypt from 1970 until 2014, respectively. Notice that the overall score of the index is provided at the top, followed by three scores corresponding to the dimensions composing the total measure of globalization: economic, social, and political globalization.

Note that in both cases, Jordan and Egypt, globalization doubled between 1970 and 2014. Economic, social, and political globalization in both Arab nations has significantly increased over the past four decades. Overall, Jordan depicts a higher level of globalization compared to Egypt. Both countries exhibit high levels of political globalization given their political history in the region: bordering Israel, being high recipients of international aid, having U.S. allies, and being active participants in the coalition against Islamist extremist groups. Essentially, the KOF scores display that more capital, information, people, ideas, services, and goods are passing in and out of Jordan and Egypt today compared to the 1970s and 1980s.

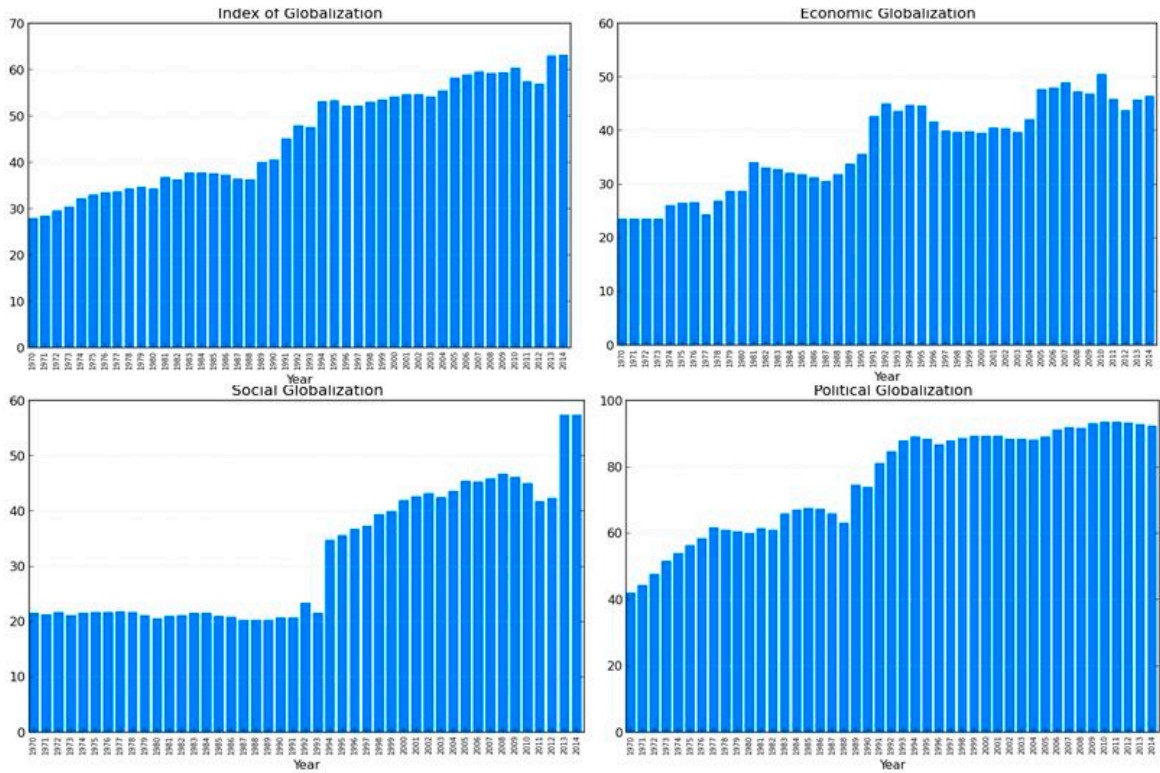

**Figure 2.** KFO globalization scores for Egypt, 1970–2014.

This paper is concerned with the configuration of cleavages in Jordan and Egypt and its transformation from the late 1980s to our contemporary period. Accordingly, two MDS analyses have been performed for each country—one for the elections of the late 1980s (Egypt) and early 1990s (Jordan), and another for a recent freely contested recent elections in 2013 (Jordan) and 2011 (Egypt). Eight total analyses, four for each country, have been conducted: two for political ideology structure and two for the parties' positions within the structure. The Inter-Parliamentary Union elections data archive was relied upon for choosing relevant parties, which were those obtaining at least two seats in the Jordanian Parliament and four seats in the Egyptian House.

To capture changes in the parties' positions, the distance between parties and issues has been computed separately for each election. The resulting structures are presented in Figure 3 through 10. The maps generated by MDS analyses may only be interpreted with respect to the distances between the objects [103–107]. They can be freely rotated for ease of interpretation, however, to preserve the originality of the results [102,106–108]. None of the maps used here have been rotated and best efforts to interpret

such complex configurations have been made. The goal is to not impose any assumptions on the structure of the dimensions or the positions of actors on issue divides. Therefore, the resulting configurations may differ with respect to countries and time periods, and careful analysis for each figure is necessary.

Figures 3 and 4 display the Jordanian political ideological structure at the political party level in 1993. Note that a single dimension was sufficient to describe the graphical configuration of objects, political parties, or political issues. Figure 3 shows the distances among relevant political parties in 1993 general elections. Notice how the Islamic Action Front occupies the furthest rightmost position, while the secular/communist-oriented Jordanian Communist Party and the Jordanian Ba'th Party occupy the left-most position on the dimensions. This clearly reflects the division on the position of religion in society, the Islamic–Secular ideological divide. This indicates that the farther one moves on the right of this dimension, the more culturally conservative one becomes. The positioning of the National Unity Party and the National Constitution Party, between the secular and Islamist actors, indicates their middle ground on the cultural divide. This result seems consistent with the conclusions in the literature [26,33,34,88] concerning political actors' discourse in the country in the early 1990s, when the political arena was dominated by the place of Islam in the newly liberal political era. In this era, the IAF, which won a relative majority seats in the 1989 elections, constituted the main political opposition to the government.

Figure 4 displays the configuration of the main issues contested in the 1993 elections in the Kingdom of Jordan. Note how welfare and budget fall to the left of the single dimension, and economic liberalism falls to the right. This is likely a representation between the contrasting visions of the Islamic Action Front, the Jordanian Communist Party, and the Jordanian Ba'th Party which called for a welfare-oriented state, whereas the National Unity Party and the National Constitution Party advocated for the economic liberalization agenda initiated by the government. Simultaneously, religious influence and sharia are found on the opposite end of cultural liberalism, pointing to the translation of the Islamic–secular divide. This dimension in Figure 4 represents the classic Islamic–secular divide where on the one end actors' call for a more religious and economically conservative state, with the other end possessing actors calling for a culturally and economically open polity.

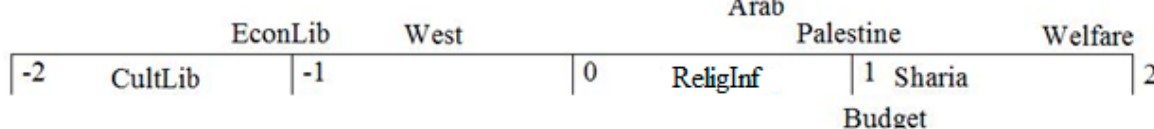

**Figure 3.** Jordanian political structure with respect to issues, 1993. EconLib: economic liberalism; CultLib: cultural liberalism; ReligInf: religious influence.

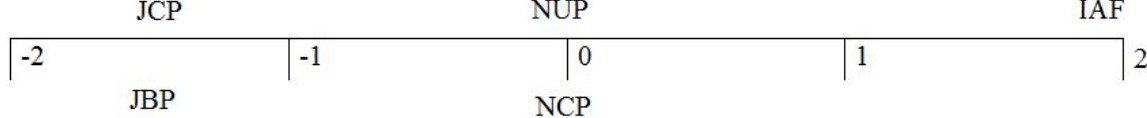

**Figure 4.** Ideological map for Jordanian parties, 1993. JCP: Jordanian Communist Party; JBP: Jordanian Ba'th Party; NUP: National Union Party; NCP: National Constitution Party; IAF: Islamic Action Front.

Figures 5 and 6 display Jordanian political ideological structures in the aftermath of the general elections held in 2013. MDS maps indicate that two dimensions were needed to sufficiently describe the configuration of objects meaningfully. It should be noted that cultural liberalism is situated at relatively opposite end of religious influence on one axis and welfare and budget are in the opposite directions of economic liberalism. This indicates a clear two-dimensional structure where economic and cultural ideological divides organized the ideologies of Jordanian political actors in 2013. Similarly, Figure 6 indicates that the Islamic Action Front and the Islamist Center Party are on opposite ends from the National Union Party, the Stronger Jordan Party, and liberal parties like the Wifaq. This is likely to represent the conflict along the role of Islam in politics and society. The placement of the Jordanian Communist Party and the Jordanian Ba'th Party at the bottom end of the second axis, while the

National Constitution Party and the National Union Party are placed on the top of the axis, represents the economic dimension division between those forces calling for a welfare state and those defending the neoliberal economic order led by the government.

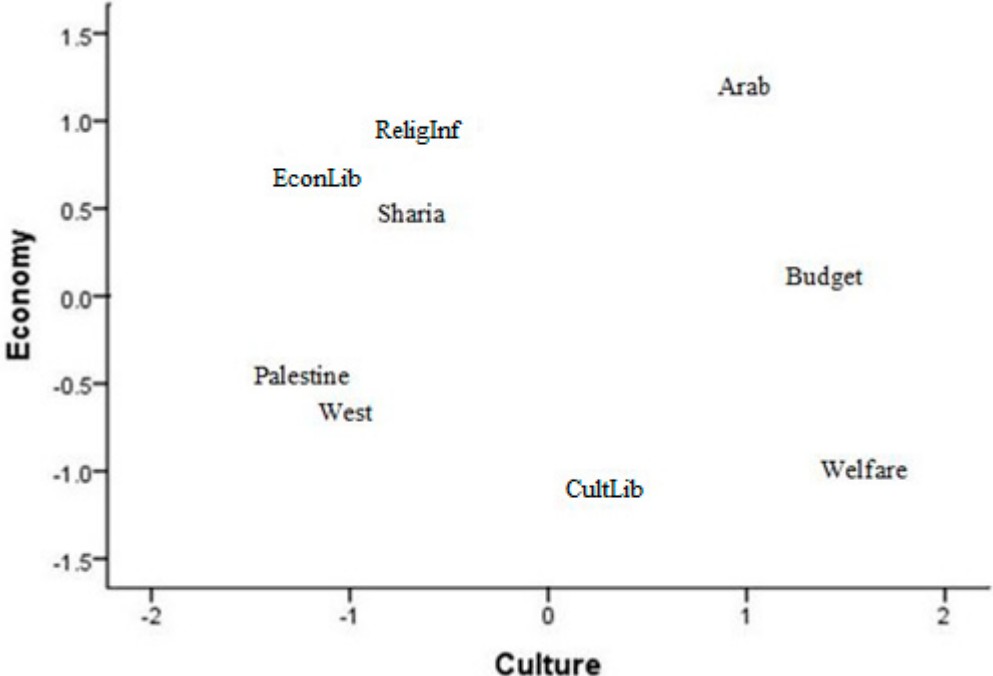

**Figure 5.** Jordanian political structures with respect to issues, 2013.

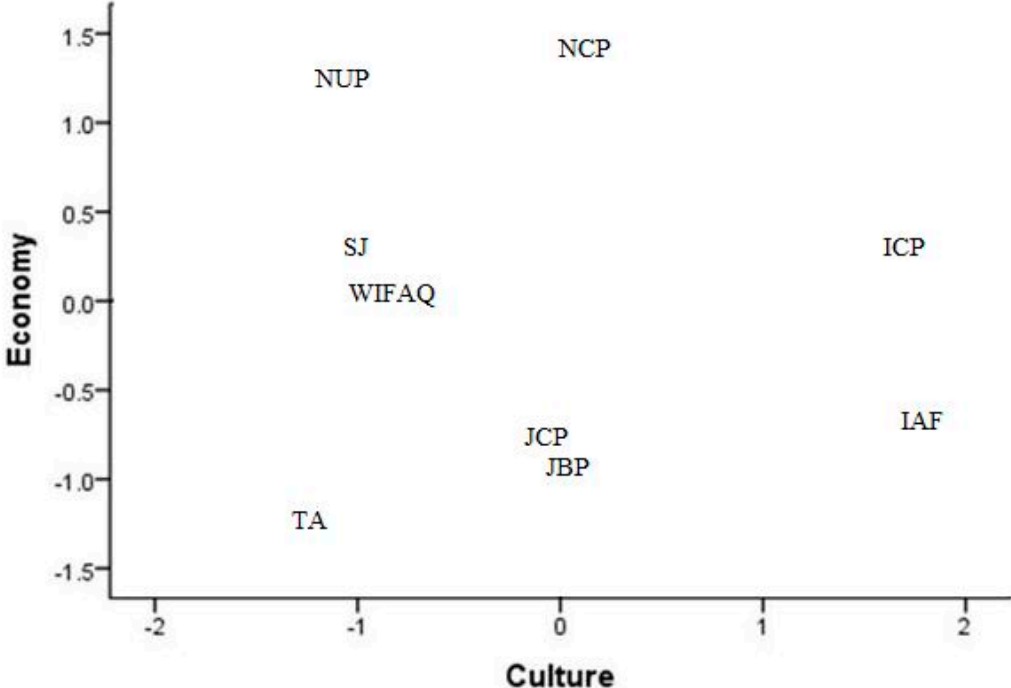

**Figure 6.** Ideological map of Jordanian parties, 2013. TA: National Current Party; SJ: Stronger Jordan; ICP: Islamic Center Party.

Figures 7 and 8 display the Egyptian political ideological structure on the party level in 1987. Figure 7 shows the configuration of political parties that won seats in the parliament during general elections that year. Notice that the Muslim Brotherhood occupies the least number of positions while liberal parties like

Ahrar, Labor, and the NPUP are on the right. At the same time, the New Wafd and the NDP are in the middle of this dimension. This is likely to represent the Islamic–secular divide where Islamic parties like the MB are found to be on one end and liberal parties like Ahrar and Labor are to be found on the other. The NDP and New Wafd have taken more centrist positions on the role of Islam in organizing the Egyptian society and politics. Figure 8 displays the distribution of political issues contested upon by parties during the 1987 elections. The figure indicates that budget, welfare, sharia, and religious influence are at one end of the dimension while economic liberalism and cultural liberalism fall on the other end. This indicates the Islamic–secular divide where on one hand Islamist actors advocate for a more culturally conservative and economically welfare vision and the other side of spectrum has parties or actors calling for religion to be out of politics and society and a more liberal economy.

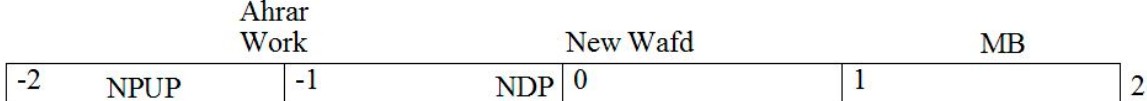

**Figure 7.** Ideological map of Egyptian parties, 1987. MB: Muslim Brotherhood.

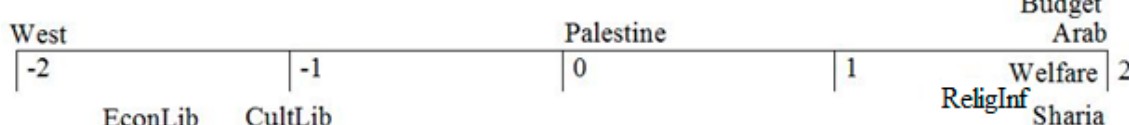

**Figure 8.** Ideological structure of Egyptian issues, 1987.

Figures 9 and 10 display the Egyptian political ideological structure in the aftermath of the 25 January Revolution of 2011. Both figures indicate that Egypt depicts a two-dimensional ideological structure with an economic and cultural component. One can note that economic liberalism is situated on one end while welfare is on the other. At the same time, religious influence and the introduction of sharia into the Egyptian social fabric is situated on one end of the axis while cultural liberalism occupies the exact opposite of the same axis. This indicates the existence of the cultural divide, the Islamic–secular conflict.

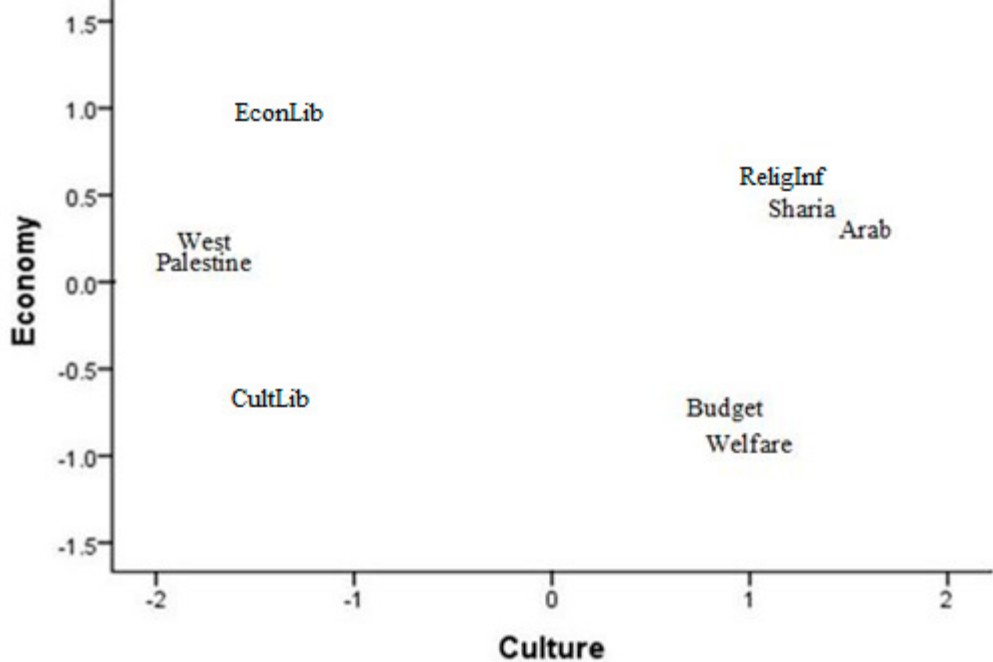

**Figure 9.** Ideological structure of Egyptian issues, 2011.

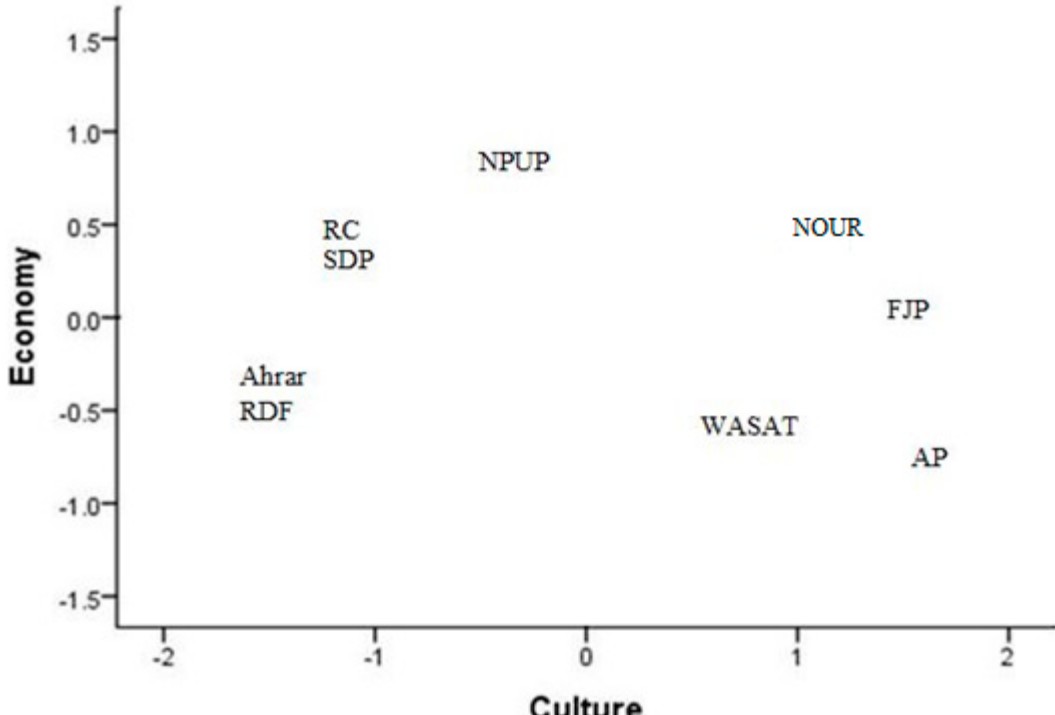

**Figure 10.** Ideological map of Egyptian parties, 2011. Al-Nour Party, RC: Revolution Continues; SDP: Social Democratic Party; Wasat; FJP: Freedom and Justice Party; AP: Authenticity Party.

Kriesi (2012) noted that the emergence of new political actors in an electorate or a transformation of established parties constitutes a significant force in changing the overall political ideological structure [39]. The emergence of Islamic centrist parties characterizes both the Jordanian and Egyptian political spaces. In the most recent parliamentary elections in Jordan and Egypt, Islamic centrist parties received a considerable share of votes (Yildirim, 2010); Islamic centrists clearly did not belong to the established Islamic opposition or the more fundamentalist elements, and were qualitatively different [109–115]. The emergence of Islamic centrist parties may reflect a conspicuous transformation of the relevant political actors and their ideological appeals in Jordan and Egypt. As they compete with mainstream and fundamentalist Islamic forces, the latter may shift policy positions to retain support within the electorate.

While Jordan and Egypt exhibit similar changes in the structure of mainstream Islamic parties (the Muslim Brotherhood), they differ with respect to the transformation of Islamic fundamentalist forces. Both countries possess populist Islamist fundamentalist groups, mostly represented by the Salafi front that explicitly challenged the Hashemite, as well as the Egyptian, regimes repeatedly. Nevertheless, the recent emergence of highly organized political parties espousing such agendas marks a departure from the earlier periods. The case of the Al-Nour Party (Party of Light), which emerged as the second victor in the first free and fair Egyptian parliamentary elections in 2011, illustrates the transformation of the radical Islamic forces in Egypt [116]. While the Al-Nour Party is authoritarian in the cultural dimension, its economic program is less obvious, with a neoliberal imprint.

## 5. Conclusions and Discussion

This section indicates that the ideological structure in Jordan and Egypt has shifted from a unidimensional structure in the 1980s to a multi-dimensional structure in the contemporary period. Throughout this time, globalization in its all forms (economic, social, and political) has significantly increased, creating new spaces of competition at all levels in Middle Eastern societies. New political actors such as the Islamist Center Parties and supporters have emerged as new winners of globalization

and developed a distinct ideological program, differentiating them from traditional Islamist parties and calling for a welfare state and a conservative cultural outlook.

A second feature is the transformation of the Islamist political camp from a highly organized political actor, the MB, to a variety of influential actors among the Islamic fundamentalist and Islamic Center Parties. In both Jordan's and Egypt's case, the Islamic Center Parties have emerged recently as main actors on the political arena with distinct positions differentiating them from the MB and its fronts (the Islamic Action Front in Jordan and the Freedom and Justice Party in Egypt). Moreover, the emergence of the Al Nour Party in Egypt presents an important development in Middle Eastern politics, where the militant Islamic fundamentalist camp agreed to play by the accepted rules and obtained significant backing from the public.

Third, political liberalization in Jordan and Egypt opened the doors for new actors to operate within the political arena. The number of political parties increased significantly in these two polities in the recent elections. This increase in parties exhibits a simultaneous increase in programmatic divergence among party groups. One noticeable development is the rise of reform movements such as Hirak in Jordan and revolutionary elements in Egypt. Both movements, represented by Stronger Jordan and Wifaq in Jordan, and the RC in Egypt, espouse a secular civil idea of the nation state. These reform movements typically exhibit liberal views on culture and variant economic outlooks.

The persistence of the cultural dimension in organizing Arab politics is likely due to the continued perception of existential threats as argued by Ingelhart [34]. The more a nation is defined by existential threats, national security affairs, religious conflicts, and identity crises, the more the prevailing political division will be along cultural terms, and in the Arab world this concerns the place of Islam in politics. The West is less defined by an existential threat, exhibiting more concern with economic or even post-materialist affairs in some cases. The continued existence of these existential crises is expected to render the religious dimension more salient in Arab politics.

It is suggested that the cultural, economic, and political denationalization trend in the Arab world has led to a transformation in the national political composition. Prior to the intensification of globalization in the 1990s and 2000s in Jordan and Egypt, three main political actors were present: (1) the governing coalition and its supporters, namely the monarchy and royalist forces in Jordan and loyal parties in the 1993 elections and the National Democratic Party in Egypt, (2) the mainstream Islamic opposition represented by the Muslim Brotherhood in both countries, as well as the Islamic Action Front, the political wing of the group, in Jordan, and (3) pan-Arab, and communist parties represented by the Ba'th in Jordan and left-wing forces in Egypt like Al-Ahrar (liberals) and the Labor Party. The most recent elections, in 2011 in Egypt and in 2013 in Jordan, also produced a newcomer, the Islamic Center Party, or as others call them, the Muslim Democrats. Winning small–medium sized enterprises that espouse a global economic and political outlook with a conservative social imprint have been advocating for a balanced neoliberal change where economic competition is in place, cultural integration is intensified regionally and globally, and religious influence still infuses societal values. The emergence of Islamic Center Parties is only one of the multifarious developments in the composition of national political spaces in the Arab world.

Sisi's military coup in Egypt has several important effects on the future of Egyptian policy expression with respect to the findings of this study. First, the state has already persecuted the Muslim Brotherhood and all political opposition parties allied with the movement, suppressing the messaging of the group. Second, the state has elevated levels of political persecution above and beyond Mubarak's regime, which was toppled in 2011. This repression indeed will narrow the political expression space, making it difficult to express alternative political visions, especially those contradicting the state's standing. Nevertheless, political expression, as argued above, will still remain, regardless of political opposition. This expression takes the format of electronic publications, sponsored documentation or any form of speech expressed by the opposition or any party for that matter.

Sisi's economic and cultural policies seem to be in favor of more globalization on all fronts. This will accelerate the effects of globalization on political preferences through the creation of winners

and losers in all segments of society. Sisi's absolute control over foreign policy-making, where alliances are made with Saudi Arabia, the United States, and Israel, is expected to increase support for the winners and disdain from losers, helping political cleavages better crystalize in the media of political expression. This argument is supported by the increasing number of political satire shows against Sisi's control. Satire with respect to Sisi and his government has become popular content across all media outlets. On TV, Aljazeera launched Above Authority in 2016, presented by the Lebanese satirist Nazih Al-Ahdab, featuring a piece on Sisi almost in every episode [117]. On YouTube, Youssef Husein, an online satirist, devoted his online show Joe Tube to making Sisi's failures sustaining material for mass-produced comedy. The satire, indeed, has begun by following Bassem Youssef's satirical program El Bernameg (The Program), copied after the Daily Show from the United States [118,119]. Sisi's government had cracked down on opposition; therefore, all such shows have been produced abroad [119]. The expression of opposition members who made it to Qatar and Turkey and the Egyptian elements was supported by the international community. Polarization on every political issue is likely to increase under Sisi's reign.

It is believed that globalization has changed existing cleavages structures in the Arab world by causing political ideological shifts in attitudes through bringing about new groups, preferences, and political/economic opportunities. This is in line with the recent theoretical developments in modernization delineated by Ingelhart and Norris [34,46,120,121]. Societies change their perceptions on culture based on their levels of modernization. While culture still dominates Arab politics, the rise in globalization has led to the emergence of a new economic conflict between political actors, as well as ordinary citizens. While neoliberalism has left many Arabs without jobs, a new family of Islamist parties has come to existence due to the economic, cultural, and political gains from globalization. Thus, economic issues are of great importance to people of the Middle East, in a similar vein to the cultural issues along the Islamist–secular division.

**Author Contributions:** Conceptualization, M.A. and I.K.; Methodology, M.A.; Software, I.K.; Validation, M.A. and I.K.; Formal Analysis, M.A.; Investigation, I.K.; References, I.K.; Writing-Original Drat Preparation, M.A.; Writing-Review & Editing, M.A.; Visualization, I.K.; Supervision, M.A.; Project Administration, M.A.; Funding Acquisition, M.A. and I.K.

**Funding:** This research received no external funding.

**Acknowledgments:** We would like to thank C. for his administrative and technical support for this project.

**Conflicts of Interest:** The authors declare no conflict of interest.

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
