# Peer review of "Globalization and the Transformation of Political Attitude Structures at the Party Level in the Arab World: Insights from the Cases of Egypt and Jordan"

_societies, doi:10.3390/soc9010024_

Round 1
Reviewer 1 Report
- While the research offers an interesting study of globalization's impact on political parties' policies, I have several reservations and concerns, which I will list below:
- I am still quite unclear on the rational or the "so what?" of this study. Is it simply because it has not been done before or is this especially significant at this point in time? If it is the latter, then, perhaps the author(s) can explain the political significance in some detail.
- Some of the writing needs to be clarified or reworded for clarity and/or grammatical and/or verb tense errors. So please be sure to edit the entire manuscript for such issues. For example, p. 6, globalizations’ should be globalization's. Even if used in the plural sense, it is normally still globalization and not globalizations.
- Another example, see this sentence in the introduction: "More recently, however, an emerging literature in comparative politics has noted that the great political impact of globalization on the domestic or national levels is large."
- Also, The Section on p. 2 from Lipset & Rokkan is repetitive in explaining dimensions are known as cleavages.
- Explain clearly what you mean by "the issue divides structure." You repeat this throughout, yet I am not quite clear on what you mean here.
- Section 2, under the subheading 2. A Theoretical Note on the Influence of Globalization on Cleavage Structures, starts off with quite a choppy paragraph that can benefit from rewriting for the purpose of clarity.
- Surely, you have evidence to back this segment, so why not cite it here? "While 98 one may argue that local elites desired to protect their regimes and opted to invite international 99 institutions to fix their economies, this logic stands weak against the ample qualitative and 100 quantitative evidence confirming the fact that Western institutions compelled the Jordanian, 101 Egyptian, Algerian, Moroccan and Algerian regimes to implement their economic and cultural 102 agendas entailing a more open market and cultural sphere."
- In your research introduction and later in the Research Design section, you claim that a number of Arab states are analyzed, then you mention Egypt and Jordan. A number, implies multiple, whereas you are analyzing two countries, so reword this to simply clarify that your research examines/analyzes Egypt and Jordan, as examples of two Arab states....
- In your methodology, you say "Party manifestos and media reports of party positions prior to elections have been taken as 260 objective sources for measuring parties’ positions on political content." How can you be sure that these are "objective"? What media outlets are you relying on? Media in the Arab World is largely state-controlled. So, are you relying on state-owned news coverage or independent newspapers? This is very important and is something that needs to be clearly explained in your study.
-Even though it is not part of your study, but it would be interesting, at least in your conclusion, to discuss the potential impact of current political military control in Egypt under Sisi, will impact your findings. The country went form political party liberalization post the Arab Spring to almost zero parties, and especially with the banning of the Muslim Brotherhood.,
Author Response
We would like to thank you for your valuable comments. We have edited the manuscript accordingly. You can find our responses to the comment in the attached file.

Reviewer 2 Report
Please find my comments in balloons in your document.
With my best wishes & regards.

Author Response
We would like to thank you for your valuable comments. We have edited the manuscript accordingly. You can find our responses to the comments in the attached file.

Round 2
Reviewer 1 Report
The manuscript reads a lot stronger now.
I see few errors in the conclusion that require editing:
You probably want to take out "It might be useful to discuss the potential impacts of current political military control in Egypt 575 under Sisi..." as you probably pasted that by mistake from your reviewer comments.
You claim that "This argument is supported by the increasing number of political satire shows against 591 Sisi’s control" Are you referring to television satire shows or online YouTube videos? You must clarify. With regards to TV shows, this is not true. There was a single satire show with Bassem Youssef the host, modeling the US The Daily Show. However, Youssef was taken to court and is currently residing in the United States after his show was banned and his was at risk of imprisonment. So please check this segment or if you have evidence of otherwise, then you must add your citations.
Also the spelling of persecuted is wrong.
Regime toppled not tappled.
on all fronts (spelling of fronts)
media is the plural of medium, so there is not such thing as mediums
Author Response
We would like to thank you for your valuable and constructive comments. We edited the manuscript accordingly.

Reviewer 2 Report
Thank your for taking into account my comments into account, the justification of your choices and the improvement of your conclusions.
NB: some typos in the new paragraphs at the end of the article
-->
579: tappled --> toppled
582: regadless --> regardless
585: fornts --> fronts
All the best!
Author Response

(The authors gave the same response as above.)
